# Synergistic Effects of Inflammation and Atherogenic Dyslipidemia on Subclinical Carotid Atherosclerosis Assessed by Ultrasound in Patients with Familial Hypercholesterolemia and Their Family Members

**DOI:** 10.3390/biomedicines10020367

**Published:** 2022-02-02

**Authors:** Po-Chih Lin, Chung-Yen Chen, Charlene Wu, Ta-Chen Su

**Affiliations:** 1Division of Cardiology, Department of Internal Medicine, National Taiwan University Hospital, Taipei 100225, Taiwan; juipeter@ntuh.gov.tw; 2Department of Environmental and Occupational Medicine, National Taiwan University Hospital, Taipei 100225, Taiwan; d09852003@ntu.edu.tw; 3Institute of Environmental and Occupational Health Sciences, College of Public Health, National Taiwan University, Taipei 100225, Taiwan; 4Global Health Program, College of Public Health, National Taiwan University, Taipei 10055, Taiwan; charlenewu@ntu.edu.tw; 5The Experimental Forest, National Taiwan University, Nantou 557009, Taiwan

**Keywords:** atherosclerosis, C-reactive protein, CRP, carotid intima-media thickness

## Abstract

Low-density lipoprotein cholesterol (LDL-C) and total to high-density lipoprotein cholesterol (TC/HDL-C) ratio are both common risk factors for atherosclerotic cardiovascular diseases (ASCVDs). However, whether high-sensitivity C-reactive protein (hsCRP) has synergistic or attenuated effects on atherogenic dyslipidemia remains unclear. We investigated subclinical carotid atherosclerosis in patients with familial hypercholesterolemia (FH) and their family members. A total of 100 families with 761 participants were prospectively studied. Participants were categorized into four groups according to atherogenic dyslipidemia and inflammatory biomarkers. The group with LDL-C ≥ 160 mg/dL (or TC/HDL-C ratio ≥ 5) combined with hsCRP ≥ 2 mg/L have a thicker carotid intima-media thickness (CIMT) in different common carotid artery (CCA) areas and a higher percentage of high plaque scores compared with other subgroups. Multivariate logistic regression analysis revealed a significantly higher adjusted odds ratio (aOR) for thicker CIMT of 3.56 (95% CI: 1.56–8.16) was noted in those with concurrent LDL-C ≥ 160 mg/dL and hsCRP ≥ 2 mg/L compared with the group with concurrent LDL-C < 160 mg/dL and hsCRP < 2 mg/L. Our results demonstrated that systemic inflammation, in terms of higher hsCRP levels ≥ 2 mg/L, synergistically contributed to atherogenic dyslipidemia of higher LDL-C or a higher TC/HDL-C ratio on subclinical atherosclerosis.

## 1. Introduction

Low-density lipoprotein cholesterol (LDL-C) is a well-established risk factor for atherosclerosis. The primary mechanism of atherogenesis is the accumulation of LDL-C in the vessel wall with the subsequent development of inflammation in the affected area and the formation of an atherosclerotic plaque [1,2]. Total to high-density lipoprotein cholesterol (TC/HDL-C) ratio as a cardiovascular disease (CVD) risk factor has also gained much attention in recent years [3,4]. Compared with using an LDL cholesterol level of 130 mg/dL as the cut-off point, using a TC/HDL-C ratio of 5 was associated with superior specificity and accuracy in predicting future coronary heart disease (CHD) [3]. The TC/HDL-C ratio also reclassifies atheroma progression and MACE rates when discordant with LDL-C, non-HDL-C, and ApoB within patients [5].

Low-grade inflammation may be essential for the accelerated progression of atherosclerosis [6,7]. High-sensitivity C-reactive protein (hsCRP), an acute-phase reactant that has been used as a marker of systemic inflammation in rheumatologic disorders, is proposed to be an emerging but reliable cardiovascular risk factor [8,9]. In a study among healthy middle-aged women, the highest CVD risk was found among those with elevated levels of both fibrinogen and CRP [10,11]. Measurement of hsCRP independently predicts future vascular events and improves global risk classification, regardless of the LDL-C level [10]. In patients with high CRP and no other major risk factor, rosuvastatin therapy significantly reduced the incidence of major cardiovascular events [11]. Rizzo M et al. found a strong correlation between events and quintiles of CRP concentrations in patients with hypertension [12]. The National Lipid Association recently established that an hsCRP level > 2 mg/L, in conjunction with major ASCVD risk factors, may be hazardous for patients with dyslipidemia [13].

Carotid artery (CA) ultrasonography is a non-invasive, reproducible, inexpensive, and radiation-free screening test [14]. Measures of CIMT are emerging technologies since CIMT is a good surrogate marker for atherosclerosis [15], an independent predictor of CV risk [16,17,18,19,20,21,22,23], and the presence of carotid plaque is a strong predictor of CV events and mortality [22,23,24]. Previous literature indicated that CIMT increased with increasing LDL-C and decreased with increasing HDL-C [25]. A TC/HDL-C ratio ≤ 5.0 was associated with a lower adjusted IMT compared to a TC/HDL-C ratio > 5 [26]. CRP can predict the progression of early carotid atherosclerosis in patients with mild to moderate cardiovascular risk and/or middle-aged patients [27]. Serum hsCRP is also correlated to CIMT in type 2 diabetic subjects [28].

This study aimed to test the hypothesis of whether inflammation and atherogenic dyslipidemia have synergistic effects on subclinical atherosclerosis in patients of FH and their family members. Patients with FH and their family members have a higher risk of premature atherosclerosis [29]. In our previous studies, patients with FH have also reported higher perceived depression [30]. There is an emergent need to identify who are at high risk of atherosclerosis in patients with FH and their families. Thus, we invited phenotypic FH patients and their families to attend this study by recording the measurement of subclinical atherosclerosis, carotid atherosclerosis, and the inflammation biomarker of hsCRP.

## 2. Materials and Methods

### 2.1. Study Design and Study Population

Details of this cohort study have been published previously [31]. We prospectively recruited 100 families totaling 761 participants from the Lipid Clinic at National Taiwan University Hospital (NTUH). Inclusion criteria of FH were at least two family members with severe hypercholesterolemia defined as total cholesterol (CHO) ≥ 290 mg/dL and LDL-C ≥ 190 mg/dL. In the FH cascade genetic study, 70% of family members were identified with mutations at LDLR.

The study was approved by the Institutional Review Board of NTUH, and all of the subjects gave their informed consent. The procedure of collecting data was standardized and dutifully followed by the physicians and assistants in measuring the variables. Information on the personal and family history of diseases, including diabetes mellitus and hypertension, was self-reported by each participant and recorded by trained assistants using a structured questionnaire. The patients’ socio-demographic characteristics, lifestyles, and history of hospitalization were also collected. Regular exercise habit was defined as the participants undertaking daily sports and leisure physical activity. Blood pressure was measured after resting for 10 min, with the subjects in the sitting position, while body weight was measured using a calibrated balance. Body mass index was calculated as weight (kilogram) divided by height (meter) squared. The circumference of the smallest part of the waist and the thickest part of the hip in the standing position was also measured.

### 2.2. Laboratory Tests

Venous blood was drawn from all participants, and blood lipid data were analyzed. Overnight fasting (>12 h) blood samples were collected for measurements of glucose, total cholesterol, HDL-C, LDL-C, and triglyceride by standard enzymatic methods with an automatic multi-channel chemical analyzer (Hitachi 7450, Hitachi Corp., Tokyo, Japan) in the central laboratory of NTUH. Serum hsCRP was measured using a chemiluminescent enzyme-labeled immunometric assay (Immulite C-Reactive Protein, Diagnostic Products Co., Los Angeles, CA, USA).

### 2.3. Extracranial Carotid Artery Ultrasound Measurements

The protocol and methods of CIMT measurements have been reported previously [32,33,34]. To summarize, a Hewlett-Packard SONO 4500 ultrasound system (Andover, MA, USA) with a 3–11 MHz real-time B-mode scanner was used for evaluation, which included observation of the longitudinal and transverse views of the extracranial carotid artery (ECCA) bilaterally. An experienced ultrasonographer performed carotid ultrasonography while the patient was supine with the neck extended in a mild lateral rotation. The carotid end-organ disease was assessed by maximal IMT at the common carotid artery (CCA) and by ECCA plaque score. Two measurements of maximal IMT at CCA 0–20 mm proximal to the carotid bifurcation were obtained bilaterally. All scans were recorded on super-VHS videotape for future and subsequent off-line analysis. Observers were blinded to the participants’ health status and risk factors. Intra-class correlation coefficients of intra-observer were about 0.70–0.87 for both sides of CCA IMT measurements, as reported previously [35].

The method for quantifying plaque score has been described previously [32,35]. Briefly, focal thickening of IMT with >50% of thickness than the adjacent IMT was considered an atherosclerotic plaque. A grade was assigned for each chosen segment: grade 0 for normal or no observable plaque; grade 1 for a small plaque with diameter stenosis <30%; grade 2 for a medium plaque with 30–49% diameter stenosis or multiple small plaques; grade 3 for one large plaque with 50–99% diameter stenosis or multiple plaques with at least one medium plaque; and grade 4 for 100% occlusion.

### 2.4. Statistical Analysis

Data were expressed as mean ± standard deviation for continuous variables, and number (percentage) if data was binary in categories. Participants were categorized into 4 groups according to atherogenic dyslipidemia and inflammation biomarker (hsCRP), in terms of LDL-C levels (≥ or <160 mg/dL) or TC/HDL-C ratio (≥ or <5), and hsCRP level (≥ or <2 mg/L). Because of this association with atherogenic dyslipidemia, CIMT, and carotid plaque score, we divided all subjects into four groups according to the distribution of hsCRP ≥ 2 mg/L or not. A trend test was used to test the significant level for the four groups. The synergistic effects of hsCRP and atherogenic dyslipidemia for carotid atherosclerosis were calculated using multivariate logistic regression models after adjusting for the following covariates: age, gender, smoking habit, alcohol consumption, and systolic blood pressure and related comorbidities.

A two-way analysis of variance (ANOVA) was conducted to compare the mean differences between groups (atherogenic dyslipidemia and inflammation biomarker) and assess the interaction between the independent variables on the dependent variable (CIMT). Multivariate logistic regression analyses were applied to estimate the adjusted odds ratios (aOR) of concurrent atherogenic dyslipidemia and inflammation on subclinical atherosclerosis after adjustment for traditional cardiovascular risk factors, such as hypertension, diabetes, smoking, alcohol, age, gender, and BMI. Results were considered significant at the 5% critical level (*p* < 0.05). Relative excess risk due to interaction (RERI), attributable proportion (AP), and synergy index (SI) were calculated to evaluate additive interaction. RERI is defined as OR_(atherogenic dyslipidemia +, hsCRP_
_≥ 2 mg/L)_-OR_(atherogenic dyslipidemia +, hsCRP < 2 mg/L)_-OR_(atherogenic dyslipidemia −, hsCRP_
_≥ 2 mg/L)_ + 1, while AP equals to RERI/OR_(atherogenic dyslipidemia +, hsCRP_
_≥ 2 mg/L)_ and SI equals to (OR_(atherogenic dyslipidemia +, hsCRP_
_≥ 2 mg/L)_ − 1)/(OR_(atherogenic dyslipidemia +, hsCRP < 2 mg/L)_ + OR_(atherogenic dyslipidemia −, hsCRP_
_≥ 2 mg/L)_ − 2). Additive interaction is absent if 0 falls into the 95% CI of RERI and AP or 1 falls into the 95% CI of SI [36,37].

All statistical analyses were performed using the SAS statistical software (version 9.4, SAS Institute Inc., Cary, NC, USA).

## 3. Results

### 3.1. Description of Study Participants

A total of 761 study subjects, 70 index cases, were diagnosed with FH. Participants with higher hsCRP levels were more likely to be older and have poorer lipids profiles. As shown in Table 1, the mean age of each quartile of hsCRP level was significantly different, with no gender difference among the baseline characteristics. The highest quartile group’s TC, triglycerides, HDL-C, LDL-C, CHO/HDL-C ratio, and LDL-C/HDL-C ratio were distinguished higher than that of hsCRP < 2 mg/L (*p* < 0.01). The prevalence of hypertension, diabetes mellitus, and smoking habits increased with hsCRP levels. The blood pressure components and fasting sugar were significantly increased with hsCRP level (*p* < 0.01). There were no statistical differences in alcohol consumption habits across hsCRP quartile distribution.

### 3.2. Predictors of Carotid Atherosclerosis

Table 2 shows the mean (mm) with a standard deviation (SD) of CCA IMT in different locations and carotid atherosclerosis plaque scores across four stratification groups of LDL-C (≥ or <160 mg/dL) and hsCRP (≥ or <2 mg/L) levels. Compared to subjects with lower LDL-C and lower hsCRP levels, those with higher LDL-C and higher hsCRP levels have a thicker CIMT in different CCA areas and a significantly higher percentage of high plaque scores. Table 3 shows the mean (mm) and SD of CCA IMT in different locations and carotid atherosclerosis plaque scores across 4 stratification groups of TC/HDL-C ratio ( ≥ or < 5) and hsCRP ( ≥ or < 2 mg/L) level. Compared to subjects with lower TC/HDL-C ratio and lower hsCRP level, those with TC/HDL-C ratio ≥ 5 and hsCRP > 2 mg/L have a thicker CIMT in different CCA areas and significantly higher percentage of high plaque scores.

### 3.3. Synergistic Effects of hsCRP with Atherogenic Dyslipidemia on Thicker CIMT

In Table 4, the two-way ANOVA test shows that both LDL-C and hsCRP explain a significant amount of variation in CIMT, and the multiplicative interaction effect between LDL-C and hsCRP on CIMT was also statistically significant. Similar results were also noted when atherogenic dyslipidemia was measured in TC/HDL-C. Yet, the variation between hsCRP groups was only marginally significant, and the interaction effect between TC/HDL-C and hsCRP on CIMT was not observed.

In Table 5, multivariate logistic regression analysis shows that subjects with LDL-C ≥ 160 mg/dL have higher aORs with 2.42 than those with LDL-C < 160 mg/dL in model 1. In Model 2, the aOR in subjects with concurrent LDL-C ≥ 160 mg/dL and hsCRP ≥ 2 mg/L was 2.64 times (95% CI: 1.22–5.69) more likely to have thicker CCA IMT compared with the other three groups. In addition, a significantly higher aOR for thicker CIMT of 3.56 (1.56–8.16) was noted in those with concurrent LDL-C ≥ 160 mg/dL and hsCRP ≥ 2 mg/L compared with the group with concurrent LDL-C < 160mg/dL and hsCRP < 2mg/L.

Multivariate logistic regression analysis showed that subjects with a TC/HDL-C ratio ≥ 5 have a higher aOR with 2.68 (95% CI: 1.67–4.30) compared to those with a TC/HDL-C ratio < 5 in model 1. In Model 2, the aOR in subjects with a concurrent TC/HDL-C ratio ≥ 5 and hsCRP ≥ 2 mg/L was 2.88 (95% CI: 1.42–5.83) times more likely to have thicker CCA IMT compared with the other three groups. In addition, a significantly higher aOR for thicker CIMT of 4.22 (95% CI: 1.87–9.54) was noted in the group with a concurrent TC/HDL-C ratio ≥ 5 and hsCRP ≥ 2 mg/L compared with the group with a concurrent TC/HDL-C ratio < 5 and hsCRP < 2 mg/L.

Similar trends were found when all participants were stratified into subjects with and without FH. Still, the strength of association of hsCRP and atherogenic dyslipidemia on CIMT was attenuated, likely due to the small sample sizes of each stratum. Both FH and non-FH subjects with LDL-C ≥ 160 mg/dL or a TC/HDL-C ratio ≥ 5 still have a significantly higher aOR for subclinical atherosclerosis.

In Table 6, 0 falls into the 95% CI of RERI and AP, while 1 falls into the 95% CI of SI for both LDL-C and TC/HDL-C, the additive interaction effect between LDL-C and hsCRP on CIMT was thus absent.

## 4. Discussion

The study demonstrates a relationship between inflammation, atherogenic dyslipidemia, and atherosclerosis. Extensive evidence has indisputably shown that LDL-C is the causal risk factor in ASCVD [38,39]. Both Europe and the United States agree that LDL is the most important risk factor for abnormal blood lipids in ASCVD. Epidemiological studies in Taiwan revealed that the TC/HDL-C > 5 ratio was a better indicator of coronary atherosclerosis compared to LDL-C (130 mg/dL) [1]. In an analysis of CHD patients in National Taiwan University Hospital, it was found that the lower HDL-C value was the more important predictor [40] if the total cholesterol was less than 200 mg/dL and the triglyceride value was less than 250 mg/dL. As for the long-term follow-up study based on the population in the Chin-Shan community, it was further found that higher non-HDL-C and ApoB values were more important than traditional LDL-C in predicting the risk of CHD. When ApoB is considered with non-HDL-C, it is found that ApoB is more important than non-HDL-C. Moreover, those with a higher TC/HDL-C ratio and a higher ApoB had a significantly higher risk of CHD [2] than those at lower risk. These local series of studies show that dyslipidemia causing arteriosclerosis, including TC/HDL-C ratio, lower HDL-C value, higher ApoB, and Non-HDL-C values, are important lipid factors for coronary artery heart disease in Taiwan residents.

Inflammation has also been found to play a causal role, independent of lipoprotein levels, in the development and progression of ASCVD [41,42,43,44,45,46]. Retrospective and prospective studies have found that hsCRP elevations are associated with acute CVD events [47], and aspirin was associated with significant reductions in the risk of MI in those with the highest hsCRP levels. Recent meta-analyses also found that hsCRP is associated with CVD events and mortality [48,49]. In the PROVE IT Study, hsCRP lowering with statin therapy was associated with reduced CVD events regardless of the LDL-C lowering [50]. In the JUPITER Study, rosuvastatin therapy reduced ASCVD events in a primary prevention population with LDL-C < 130 mg/dL but an elevated hsCRP ≥ 2 mg/L [11]. Validation for the role of inflammation in ASCVD was first highlighted recently after the CANTOS (Canakinumab Anti-inflammatory Thrombosis Outcome Study) showed a reduction in recurrent CVD events with canakinumab treatment [51], especially among those who achieved hsCRP < 2 mg/L at three months on therapy [52]. However, in the CIRT study, low-dose methotrexate did not reduce levels of interleukin-1β, interleukin-6, or CRP and did not result in fewer CVD events than a placebo in patients with chronic coronary disease [53]. The LoDoCo Study showed that low-dose colchicine (0.5 mg twice daily) can effectively decrease hsCRP in patients with clinically stable CAD and increase hsCRP independent of aspirin and atorvastatin use [54]. Among patients with a recent myocardial infarction, colchicine at a dose of 0.5 mg daily led to a significantly lower risk of ischemic cardiovascular events than a placebo (COLCOT) [55]. In the LoDoCo-2 Study, colchicine (0.5 mg once daily) demonstrated considerably lower CVD events compared to a placebo in patients with chronic coronary disease in 2020 [56].

The strength of our study includes the complete measurement of CIMT, hsCRP, and lipid profiles as the focus on FH proband and their family members. Subjects of higher hsCRP also have a higher prevalence of cardiovascular risk factors also corroborated the validity of hsCRP measurements in Table 1. To our knowledge, this is the first study to prove the synergistic effect of inflammation on atherogenic dyslipidemia in FH and their families. The limitation of this study includes the following: First, this study is cross-sectional; thus, we cannot infer a causal relationship. Second, as shown in Table 1, the subjects’ age, blood pressure, fasting sugar, and the prevalence of hypertension, diabetes mellitus, and smoking habits increased along the hsCRP gradient. Although we have adjusted these factors by incorporating them as confounding factors in the logistic regression models, the study is still prone to unmeasured confounding risk factors that are simultaneously associated with hsCRP and subclinical carotid atherosclerosis. Third, information regarding infection status, such as urinary tract infection, upper respiratory infection, or chronic inflammation due to periodontitis, was not obtained. Thus, we cannot exclude patients with current infection or inflammatory disease from subjects with higher hsCRP in the multivariate regression analysis. This may lead to overinflating or underestimating the estimate in the multivariate analysis. Finally, we use carotid plaque and intima-media thickness, two biologically and genetically distinct entities representing different atherosclerosis phenotypes, to assess subclinical carotid atherosclerosis comprehensively [57]. Although CIMT is a widely used and well-established biomarker for subclinical atherosclerosis, the thickening of the intima-media is not directly equated to atherosclerosis. The study results should thus be interpreted with caution.

## 5. Conclusions

This study confirmed that systemic inflammation, in terms of higher hsCRP levels ≥ 2 mg/L, synergistically contributed to atherogenic dyslipidemia of higher LDL-C or TC/HDL-C ratio on subclinical atherosclerosis. In addition to LDL-C or TC/HDL-C ratio, measurement of hsCRP can provide additional prognostic effects for significant carotid atherosclerosis in FH and their family members. Moreover, our findings echo the guideline suggestion of hsCRP measurement in selected individuals for ASCVD primary prevention, and through control of hsCRP may benefit the prevention of ASCVD.

## Figures and Tables

**Table 1 biomedicines-10-00367-t001:** Basic characteristics of familial hypercholesterolemia by quartile of hsCRP.

	Total	hsCRP (mg/L)				*p* Value for Trend
		Q_1_ (<0.4) *n* = 192	Q_2_ (0.4–0.7) *n* = 190	Q_3_ (0.7–1.6) *n* = 190	Q_4_ (>1.6) *n* = 189	
Male, %	48.37	45.31	51.05	48.95	47.62	0.3810
Age, y/o	42.38 ± 17.98	32.66 ± 17.29	44.98 ± 16.32	46.40 ± 16.94	45.74 ± 17.91	<0.0001
Cholesterol, mg/dL	236.57 ± 66.72	221.81 ± 69.16	235.71 ± 56.79	240.58 ± 67.98	246.86 ± 68.30	0.0019
Triglyceride, mg/dL	131.05 ± 167.18	82.27 ± 39.09	114.59 ± 75.87	136.11 ± 94.17	152.63 ± 105.3	<0.0001
HDL-C, mg/dL	54.92 ± 16.67	58.73 ± 19.52	55.95 ± 14.85	54.29 ± 13.76	51.70 ± 16.23	0.0003
LDL-C, mg/dL	155.57 ± 64.67	133.37 ± 65.17	147.81 ± 52.87	150.05 ± 63.39	156.87 ± 60.91	0.0018
CHO/HDL-C	4.53 ± 1.62	3.96 ± 1.41	4.39 ± 1.21	4.65 ± 1.61	5.04 ± 1.73	<0.0001
LDL/HDL-C	2.99 ± 1.41	2.43 ± 1.32	2.79 ± 1.13	2.93 ± 1.43	3.24 ± 1.48	<0.0001
Diabetes mellitus, %	4.48	1.05	4.21	5.32	7.41	0.0014
Fasting glucose, mg/dL	92.47 ± 16.65	87.05 ± 8.08	91.18 ± 15.83	95.02 ± 14.69	96.62 ± 23.13	<0.0001
Hypertension, %	21.23	8.90	22.99	23.40	29.79	<0.0001
Systolic BP, mmHg	111.86 ± 24.80	107.21 ± 14.47	115.60 ± 18.01	117.34 ± 16.42	118.70 ± 18.50	<0.0001
Diastolic BP, mmHg	70.48 ± 15.85	67.18 ± 10.32	72.69 ± 11.05	74.20 ± 10.85	74.99 ± 11.32	<0.0001
Smoking habit, %	17.35	10.94	18.95	20.53	19.05	0.0170
Alcohol habit, %	16.01	12.63	17.65	17.02	16.76	0.1609
Carotid IMT, mm	0.57 ± 0.19	0.53 ± 0.13	0.60 ± 0.15	0.63 ± 0.20	0.62 ± 0.16	<0.0001
Plaque Score	1.92 ± 3.33	0.87 ± 1.89	2.17 ± 3.46	2.73 ± 4.16	2.18 ± 3.32	<0.0001
Statin use, %	26.41	18.75	28.42	34.74	23.81	0.1283

**Table 2 biomedicines-10-00367-t002:** Distribution of common CIMT stratified by LDL-C and hsCRP levels.

	LDL-C ≥ 160 mg/dL and hsCRP ≥ 2 mg/L (*n* = 36)	LDL-C ≥ 160 mg/dL and hsCRP < 2 mg/L (*n* = 162)	LDL-C < 160 mg/dL and hsCRP ≥ 2 mg/L (*n* = 112)	LDL-C < 160 mg/dL and hsCRP < 2 mg/L (*n* = 444)	*p* Value
IMT					
RCCA1, mm	0.65 ± 0.17	0.63 ± 0.22	0.62 ± 0.20	0.57 ± 0.16	0.0005
RCCA2, mm	0.62 ± 0.16	0.59 ± 0.18	0.60 ± 0.20	0.55 ± 0.15	0.0013
LCCA1, mm	0.70 ± 0.21	0.66 ± 0.23	0.60 ± 0.19	0.57 ± 0.16	<0.0001
LCCA2, mm	0.72 ± 0.22	0.64 ± 0.27	0.61 ± 0.18	0.57 ± 0.16	<0.0001
IMT mean, mm	0.67 ± 0.16	0.63 ± 0.20	0.61 ± 0.17	0.57 ± 0.14	<0.0001
Plaque Score	3.92 ± 4.31	3.10 ± 4.09	1.92 ± 3.09	1.45 ± 2.90	<0.0001
0%	30.56	36.42	59.56	64.33	<0.0001
1–2%	16.67	22.84	11.71	15.35	
3–5%	25.00	19.14	17.12	11.51	
≥6%	27.78	21.60	12.61	8.80	

**Table 3 biomedicines-10-00367-t003:** Distribution of Common CIMT stratified by TC/HDL-C ratio and hsCRP levels.

	TC/HDL ≥ 5 and hsCRP ≥ 2 mg/L (*n* = 70)	TC/HDL ≥ 5 and hsCRP < 2 mg/L (*n* = 172)	TC/HDL < 5 and hsCRP ≥ 2 mg/L (*n* = 78)	TC/HDL < 5 and hsCRP< 2 mg/L (*n* = 434)	*p* Value
IMT					
RCCA1, mm	0.68 ± 0.22	0.64 ± 0.19	0.58 ± 0.15	0.57 ± 0.17	<0.0001
RCCA2, mm	0.67 ± 0.21	0.59 ± 0.15	0.55 ± 0.14	0.55 ± 0.17	<0.0001
LCCA1, mm	0.67 ± 0.19	0.65 ± 0.20	0.59 ± 0.19	0.57 ± 0.17	<0.0001
LCCA2, mm	0.69 ± 0.20	0.63 ± 0.19	0.58 ± 0.17	0.57 ± 0.20	<0.0001
IMT mean, mm	0.68 ± 0.18	0.63 ± 0.16	0.58 ± 0.15	0.57 ± 0.16	<0.0001
Plaque Score	2.78 ± 3.62	2.91 ± 4.14	2.08 ± 3.41	1.48 ± 2.86	<0.0001
0%	42.03	42.20	60.26	62.73	<0.0001
1–2%	17.39	21.97	8.97	15.51	
3–5%	23.19	15.03	15.38	12.96	
≥6%	17.39	20.81	15.38	8.80	

**Table 4 biomedicines-10-00367-t004:** Two-way ANOVA test assessing effects of serum hsCRP and atherogenic dyslipidemia on subclinical atherosclerosis (IMT ≥ 75th) in patients with FH and their families.

Source	Partial SS	df	MS	F	*p* Value
LDL-C as predictor					
Model	5.64	3	1.88	10.35	0.000
LDL-C (≥160 vs. <160 mg/dL)	4.21	1	4.21	23.20	0.000
hsCRP (≥2 vs. <2 mg/L)	1.38	1	1.38	7.58	0.006
LDL-C*hsCRP	0.73	1	0.73	4.02	0.045
Residual	138.17	757	0.18		
Total	143.81	760			
R-squared = 0.039; Adjusted R-squared = 0.036					
TC/HDL-C as predictor					
Model	5.59	3	1.86	10.26	0.000
TC/HDL-C (≥5 vs. < 5)	3.52	1	3.52	19.37	0.000
hsCRP (≥2 mg/L vs. <2 mg/L)	0.65	1	0.65	3.57	0.059
TC/HDL-C*hsCRP	0.05	1	0.05	0.25	0.618
Residual	138.22	757	0.18		
Total	143.81	760			
R-squared = 0.039; Adjusted R-squared = 0.035					

SS = sum of squares, df = degree of freedom, MS = mean square.

**Table 5 biomedicines-10-00367-t005:** Additional prognostic effects of serum hsCRP with atherogenic dyslipidemia as a predictor of subclinical atherosclerosis in patients with FH and their families.

	IMT ≥ 75th Percent		
	All Participants	FH Participants	Non-FH Participants
LDL-C as a predictor			
Model 1 LDL-C ≥ 160 mg/dL vs. LDL-C < 160 mg/dL	2.42(1.53–3.82) ‡	1.25(0.47–3.29)	1.86(1.04–3.31) *
Model 2 LDL-C ≥ 160 mg/dL and hsCRP ≥ 2 mg/L vs. other 3 groups	2.64(1.22–5.69) *	2.17(0.85–5.54)	1.38(0.48–3.97)
Model 3 LDL-C ≥ 160 mg/dL and hsCRP ≥ 2 mg/L vs. LDL-C < 160 mg/dL and hsCRP < 2 mg/L	3.56(1.56–8.16) ‡	1.55(0.35–6.92)	1.60(0.51–5.01)
TC/HDL-C as a predictor			
Model 1 TC/HDL-C ≥ 5 vs. TC/HDL-C < 5	2.68(1.67–4.30) ‡	1.35(0.60–3.06)	2.15(1.12–4.12) *
Model 2 TC/HDL-C ≥ 5 and hsCRP ≥ 2 mg/L vs. other 3 groups	2.88(1.42–5.83) ‡	1.69(0.66–4.29)	2.21(0.80–6.10)
Model 3 TC/HDL-C ≥ 5 and hsCRP ≥ 2 mg/L vs. TC/HDL-C < 5 and CRP< 2 mg/L	4.22(1.87–9.54) ‡	1.38(0.42–4.61)	2.56(0.81–8.09)

Data were presented after controlling age, male, hypertension, systolic BP, diastolic BP, fasting sugar, BMI, smoking, and alcohol habit. *p*-value for these parameters * *p* < 0.05, ‡ < 0.005.

**Table 6 biomedicines-10-00367-t006:** Additive interaction analysis of serum hsCRP and atherogenic dyslipidemia on subclinical atherosclerosis (IMT ≥ 75th) in patients with FH and their families.

	RERI	AP	SI
LDL-C ≥ 160 mg/dL, hsCRP ≥ 2 mg/L	1.020 (−0.992–3.032)	0.366 (−0.165,0.898)	2.333 (0.437–12.441)
TC/HDL-C ≥ 5, hsCRP ≥ 2 mg/L	1.148 (−0.880,3.175)	0.402 (−0.101,0.905)	2.629 (0.452,15.305)

RERI: relative excess risk due to interaction= OR_(atherogenic dyslipidemia +, hsCRP_
_≥ 2 mg/L)_-OR_(atherogenic dyslipidemia +, hsCRP < 2 mg/L)_-OR_(atherogenic dyslipidemia −, hsCRP_
_≥ 2 mg/L)_ + 1; AP: attributable proportion = RERI/OR_(atherogenic dyslipidemia +, hsCRP_
_≥ 2 mg/L)_; SI: synergy index = (OR_(atherogenic dyslipidemia +, hsCRP_
_≥ 2 mg/L)_ − 1)/(OR_(atherogenic dyslipidemia +, hsCRP < 2 mg/L)_+ OR_(atherogenic dyslipidemia −, hsCRP_
_≥ 2 mg/L)_ − 2).

## Data Availability

All of the relevant data are presented in this article.

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
