# Peer review of "Synergistic Effects of Inflammation and Atherogenic Dyslipidemia on Subclinical Carotid Atherosclerosis Assessed by Ultrasound in Patients with Familial Hypercholesterolemia and Their Family Members"

_biomedicines, 2022, doi:10.3390/biomedicines10020367_

Round 1
Reviewer 1 Report
In the present manuscript the authors tested the hypothesis if inflammation and atherogenic dyslipidemia have synergistic effects on subclinical atherosclerosis in FH patients and their family members.
The participant with high CRP levels were different from those with low levels in term of age (older) lipid profile (poorer) blood pressure (higher) and smoking habits (higher). Considering this baseline, the authors have found that subjects with higher LDL-C and higher hsCRP levels have a thicker CIMT in different CCA areas.
Finally, the authors found a synergistic effects of hsCRP with atherogenic dyslipidemia on thicker CIMT.
Comments
- The baseline characteristics of the studied population are significantly different along different grade of hsCRP levels. This represents a very important limitation of the study, although a multivariate regression analysis has been performed. Thus, to my opinion it is very likely that the association between hsCRP levels and CIMT has been driven by many other pro-atherosclerotic factors: lipids, blood pressure, age and smoking. The authors should comment this aspect as a very limiting factor of the study
- It is not clear how the authors have determined the “synergistic” effect of hsCRP and dyslipidemia. To my opinion we have an additive action. What type of statistical has been performed to reach this conclusion?
- Does the authors find a different association between hsCRP and dyslipidemia with CIMT between FH and not FH participants?
Author Response
Thank you very much for the important comments. We have responded to reviewers' comments and revised our manuscript in a point-to-point manner accordingly as follows.
Reviewer 1:
- The baseline characteristics of the studied population are significantly different along different grade of hsCRP levels. This represents a very important limitation of the study, although a multivariate regression analysis has been performed. Thus, to my opinion it is very likely that the association between hsCRP levels and CIMT has been driven by many other pro-atherosclerotic factors: lipids, blood pressure, age and smoking. The authors should comment this aspect as a very limiting factor of the study.
Response: Thank you for your comment. We have included an additional limitation in the discussion section for better clarification. To be precise, as shown in Table 1, the subjects' age, blood pressure, fasting sugar, and the prevalence of hypertension, diabetes mellitus, and smoking habits increased along the hsCRP gradient. We have adjusted these factors by incorporating them as confounding factors. However, the study is still prone to unmeasured confounding risk factors simultaneously associated with hsCRP and subclinical carotid atherosclerosis.
- It is not clear how the authors have determined the "synergistic" effect of hsCRP and dyslipidemia. To my opinion we have an additive action. What type of statistical has been performed to reach this conclusion?
Response: Thank you for your comment. We have performed an ANOVA test and included an interactive term (atherogenic dyslipidemia * hsCRP) in the new Table 4 to demonstrate the multiplicative interaction effects between atherogenic dyslipidemia and hsCRP. We also performed an interaction analysis to assess the additive interaction effects in the new Table 6. A detailed description was also added in the results section.
- Does the authors find a different association between hsCRP and dyslipidemia with CIMT between FH and not FH participants?
Response: We have stratified all participants into subjects with and without FH, and similar trends were found (please refer to Table 5 [previously Table 4]). Still, the strength of association of hsCRP and atherogenic dyslipidemia on CIMT was attenuated, likely due to the small sample sizes of each stratum.
Reviewer 2 Report
Authors reported a research article with the aim of elucidating subclinical carotid atherosclerosis in patients with familial hypercholesterolemia (FH) and their family members. They prospectively included 100 families with 761 participants to determine atherogenic dyslipidemia and inflammatory biomarkers. Authors found that higher hsCRP levels >2 mg/L, synergistically contributed to atherogenic dyslipidemia of higher LDL-C or higher TC/HDL-C ratio on subclinical atherosclerosis. Although these findings seem to be interesting, I would like to put forward several items to discuss.
- Authors declared that they had obcerved asymptomatic atherosclerosis, but Extracranial Carotid Artery Ultrasound Measurements was used to determined a focal thickness of carotid intima-media segment and also plaque(s). There remained to be uncertain whether this parameter would thoroughly elucidate a severity of atherosclerosis and 2021 ESC Guidelines on cardiovascular disease prevention does not recommend to provide it with this purpose. https://academic.oup.com/eurheartj/article/42/34/3227/6358713 However, authors should modify the title of the article so that the biomarker of subclinical atherosclerosis was reported.
- Please, go though the text of the article and check whether diabetes mellitus was given and T2DM. Yet, substitute DM for T2DM in Table 1.
- Please, give extensive characteristics of the patients enrolled to the study especially having comorbidities. Moreover, the methods by which authors had detemined these conditions should be reported in the section "Methods".
- Predictors of IMT (see Table 4) should be compared ech other with ANOVA.
- Section "Study limitations" requires being reported along with a clear explanation whether statistical bias on asymptomatic atherosclerosis determination by the method mentioned above could influence the final interpretation.
Author Response
Thank you very much for the important comments. We have responded to reviewers' comments and revised our manuscript in a point-to-point manner accordingly as follows.
Reviewer 2:
- Authors declared that they had observed asymptomatic atherosclerosis, but Extracranial Carotid Artery Ultrasound Measurements was used to determined a focal thickness of carotid intima-media segment and also plaque(s). There remained to be uncertain whether this parameter would thoroughly elucidate a severity of atherosclerosis and 2021 ESC Guidelines on cardiovascular disease prevention does not recommend to provide it with this purpose. https://academic.oup.com/eurheartj/article/42/34/3227/6358713 However, authors should modify the title of the article so that the biomarker of subclinical atherosclerosis was reported.
Response: Thanks for your comment. Although thickening of the intima-media is not directly equated to atherosclerosis, intima-media thickness is a widely used and early surrogate biomarker for subclinical atherosclerosis. As suggested, we modified the article title by underscoring "subclinical carotid atherosclerosis assessed by ultrasound."
- Please, go through the text of the article and check whether diabetes mellitus was given and T2DM. Yet, substitute DM for T2DM in Table 1.
Response: Thanks for your comment. Data on the personal history of diseases, including diabetes mellitus, were collected through a questionnaire. We did not distinguish Type II DM from Type I or other types of DM in the original questionnaire. Thus, we keep "diabetes mellitus" in Table 1 and the text instead of changing to "T2DM."
- Please, give extensive characteristics of the patients enrolled to the study especially having comorbidities. Moreover, the methods by which authors had determined these conditions should be reported in the section "Methods".
Response: Thanks for your comment. In Table 1, we have detailed the subjects' subjective (self-reported diabetes mellitus, hypertension, smoking and alcohol habit, etc.) and objective characteristics (blood lipids, fasting sugar, blood pressure, carotid-intima thickness, etc.). We have edited the methods section to describe methods to determine these conditions in greater detail. Information on the personal history of diseases, including diabetes mellitus and hypertension, was self-reported by each participant and recorded by trained assistants using a structured questionnaire.
- Predictors of IMT (see Table 4) should be compared each other with ANOVA.
Response: Thanks for your comment. We conducted the ANOVA in the new Table 4 and described the result in the results section.
- Section "Study limitations" requires being reported along with a clear explanation whether statistical bias on asymptomatic atherosclerosis determination by the method mentioned above could influence the final interpretation.
Response: Thanks for your comment. We use carotid plaque and intima-media thickness, two biologically and genetically distinct entities representing different phenotypes of atherosclerosis, as surrogate biomarkers for subclinical carotid atherosclerosis. Although widely used and well established, the thickening of the intima-media is not directly equated to atherosclerosis. As suggested, we have listed this as a limitation in the discussion section.
Round 2
Reviewer 1 Report
I have no further comments
Reviewer 2 Report
Authors reported a revised version of the article along with comprehensive explanation of tha in which way the corrections were made. I have no serious flaws to the article in its revised version.